# Effect of community antiretroviral therapy on treatment outcomes among stable antiretroviral therapy patients in Nigeria: A quasi experimental study

Patrick Dakum[1,2], Juliet Ajav-Nyior[1], Timothy A. Attah[1] *, Gbenga A. Kayode[1,3], Asabe Gomwalk[1], Helen Omuh[1], Halima Ibrahim[1], Mercy Omozuafoh[1], Abimiku Alash'Ie[2,3], Charles Mensah[1], Young Oluokun[1], Franca Akolawole[1]

1 Institute of Human Virology, Maina Court, Central Business District, Abuja, Nigeria, 2 Institute of Human Virology University of Maryland School of Medicine, Baltimore, Maryland, United States of America, 3 International Research Centre of Excellence, Institute of Human Virology, Maina Court, Central Business District, Abuja, Nigeria

* attah_adejoh@yahoo.com

**Data Availability Statement:** Due to the ethical concerns and the need for protection of confidentiality of research subjects, data can only

## Abstract

### Objectives

This study evaluates the effect of Community Anti-retroviral Groups on Immunologic, Virologic and clinical outcomes of stable Antiretroviral Therapy patients in Nigeria.

### Method

A cohort of 251 eligible adults ($\geq$18 years) on first-line ART for at least 6 months with CD4 counts >200 cells/mm3 and viral load <1000 c/ml were devolved from 10 healthcare facilities to 51 community antiretroviral therapy groups. Baseline immunologic, virologic and clinical parameters were collected and community antiretroviral therapy group patients were followed up for a year after which Human Immunodeficiency Virus treatment outcomes at the baseline and a year after follow-up were compared using paired sample t-test. All the analyses were performed in STATA version 14.

### Result

Out of the 251 stable antiretroviral therapy adults enrolled, 186 (75.3%) were female, 52 (22.7%) had attained post-secondary education and the mean age of participants was 38 years (SD: 9.5). Also, 66 (27.9%) were employed while 125 (52.7%) were self-employed and 46(19.41%) unemployed. 246 (98.0%) of the participants were retained in care. While there was no statistically significant change in the CD4 counts (456cells/mm3 vs 481cells/mm3 P-0.489) and Log$_{10}$ viral load (3.54c/ml vs 3.69c/ml P-0.359) after one year of devolvement into the community, we observed a significant increase in body weight (60.8 vs 65, P-0.01).

be shared upon approval by the Institute of Human Virology Nigeria. To gain access to this data, an official request must be submitted to the institute provided it meets the recommended ethical requirement. Qualified, interested researchers may contact: Eunice Ekong, Retention (Strategic Information), eekong@ihvnigeria.org.

**Funding:** The authors received no specific funding for this work.

**Competing interests:** The authors have declared that competing interest exist.

## Conclusion

This study demonstrates that community antiretroviral therapy has a potential of maintaining optimum treatment outcomes while improving adherence and retention, and reducing the burden of HIV treatment on the health facility. This study provides baseline information for further research and vital information for HIV program implementers planning to decentralize the management of stable antiretroviral therapy clients.

## Introduction

The advent of Antiretroviral Therapy (ART) in Nigeria has long played a momentous role in improving the quality of care among People Living with Human Immunodeficiency Virus (PLHIV) [1–3]. This involves the use of a combination of Antiretroviral (ARV) drugs in the management of the disease, prevention of co-morbidities and progression to Acquired Immunodeficiency Syndrome (AIDS) and AIDS related mortalities. However, ART is most effective when adherence and retention in care are optimum [4–6].

Between 2012 and 2016, there was a 16.9% decline in the burden of AIDS-related deaths in Nigeria to the advent and scale up of ART [7]. As scale-up of ART programs continues to increase the number of PLHIV on care in Sub-Saharan Africa (SSA), one of the most significant challenges is the heavy burden on the health system [8–10]. This has negatively affected the capacity of the health workforce to function efficiently in providing quality care for PLHIV [8–10]. Regionally, there is a substantial deficit in the number of health workers. The Nigerian health workforce remains relatively below (8.7% clinician deficit) the recommended 23 clinicians per 1,000 population [11–13]. There is also an increased risk of new HIV infections among health care workers of which health support staff are considered to be most vulnerable [12–17]. As a result, strategies have been adopted and deployed to meet the high patient demand and also mitigate the risk of nosocomial infections [11–13].

As the global community strives towards achieving the Joint United Nations Programme on HIV/AIDS (UNAIDS) 90-90-90 ambitious targets, the World Health Organization (WHO) has recommended that ART be decentralized for stable ART patients in low- and middle-income countries [18]. Therefore, countries supported by the President's Emergency Plan for AIDS Relief (PEPFAR) have taken significant steps to implement these strategies of HIV treatment and care for stable ART patients [19–21]. While controversies about the cost effectiveness and intricacies of ART treatment (adherence, logistics and resistance to ARVs)in SSA countries exist, the implementation of a decentralized ART model is inevitable as this may be significant in achieving the ambitious 90-90-90 regionally [22,23].

It is estimated that 95% of HIV service delivery is health facility-based in low and middle income countries [24]. In nearly all countries, delivery of HIV care in the initial phase of rapid scale-up has been based on a "one-size-fits-all", clinic-based approach. Community ART (cART) is a HIV treatment model which decentralizes ART from the healthcare facilities into the community. CAG consists of a group of self-formed stable ART clients who take turns in attending clinical assessment and monitoring tests at the health facility, whilst representing other group members in routine drug pickup [25]. This model fosters the scale up of ART program for stable clients at the community level using a task-shifted ART delivery approach in order to increase clients' participation and empowerment in care [26,27]. Also, CAG presents potential benefits in providing easy access to ART by addressing the financial and time constraint associated with frequent clinic visits. Therefore, this model has the potential of

improving retention in care and adherence to treatment while providing stronger community engagement, social network and accountability mechanisms towards the health system through advocacy for optimum HIV service delivery. The CAG model also fosters patient self-management and independence in treatment [25].

In SSA, previous studies have shown significant improvements in clinical, virologic and immunologic outcomes among stable ART clients upon the initiation of a decentralized model of care [28,29]. Improved retention in care and self reported adherence have also been identified among stable ART patients devolved into CAGs [28,29]. However, one study conducted in South Africa highlighted concerns about stigma, confidentiality and poor interaction with healthcare workers which may be identified as one of the limitations of the model [30].

Based on recommendations from the WHO, Nigeria adopted and piloted CAG model for stable ART patients in 2017. However, no study has been conducted to examine the effect of this model on treatment outcomes. Thus, this study evaluates the effect of CAGs on clinical, virologic and immunologic outcomes of stable ART patients in Nigeria.

## Methods

### Study design and sampling

Adopting a quasi-experimental approach, a pilot study was conducted on 251 stable ART patients who were enrolled into 51 CAGs to examine differences in the clinical, immunologic and virologic status of the patients, before and one year after devolvement of patients from the healthcare facilities, into the community.

### Study setting

Nigeria is the most populous African country with a population of over 193 million people [31]. Currently, the national prevalence of HIV is 1.4% giving a population of about 1.9 million PLHIV and only 33% on ART [32,33]. In 2001, Nigeria commenced ART scale-up with only 25 tertiary health facilities [34]. Although significant progress has been achieved with about 7075 HIV counseling and Treatment (HCT) sites in 2013, these sites are however insufficient to efficiently identify and manage the total population of ART patients in Nigeria. Also, most of these sites are challenged with poor health infrastructure as well as shortage of health care professionals [7]. As a result, HIV programs have been recommended for decentralization into the communities in order to ensure easy and timely access to ART, reduce the workload on healthcare facilities and improve retention and adherence to care among stable PLHIV.

### Study sites and participants

Out of 11 Institute of Human Virology, Nigeria (IHVN) supported health care facilities, 251 eligible HIV clients were devolved into 51 CAGs. cART was initiated in August 2017 and treatment outcomes were assessed after a one year follow-up period.

### Eligibility criteria

ART patients who were $\geq$18 years, on first-line ART regimen for $\geq$ 6 months, had a current CD4 count > 200cells/mm$^3$ as well as a viral load <1000c/ml, had neither a current opportunistic infection nor co-morbidity and had made an informed decision to be part of a CAG at the start of the study were recruited and integrated into the study.

## Community ART groups

CAG are self-formed groups consisting of 4–8 stable ART patients living within the same community. CAG model considers stable ART patients as the centre of health care delivery. The 4 key functions of CAG model include the facilitation of monthly ART distribution to members within the same group, provision of adherence and social support, monitoring of wellbeing of CAG members and ensuring rotational clinical consultation for each CAG member at least once every 6 months.

During group meetings, a representative is expected to distribute ARVs to each member of the group, and a peer counselor provides adherence support. Clinical consultations and investigations are also conducted at the clinic as members of the group take turns in collecting ARV refills. However, in situations where CAG members are presented with serious health complaints, members are advised to visit the health facility for prompt and adequate care.

## Data collection and study variables

Demographic, socioeconomic and health information of CAG members were retrieved from the Open Medical Record System (OMRS) at the health facilities. Data retrieved from OMRS included participants' age, sex, education, occupation, date of ART initiation, ARV pickup history, ART regimen, adherence, baseline viral load, CD4 and body weight, viral load, CD4 and body weight after a year follow-up, opportunistic infections and LTFU. Retention in care was defined as any stable ART patient who remained within their CAGs (without default in routine clinic visits) at the end of the one year follow up period while LTFU was defined as any CAG member who failed to attend expected clinical visit for drug pick up, routine consultation as well as laboratory investigations.

## Data analysis

Data retrieved from patients records were transcribed into a Microsoft Excel spreadsheet. After data cleaning and coding, descriptive statistics was performed using STATA version 14. Categorical variables were presented as numbers and percentages, while continuous data were expressed using mean and standard deviation. Participants' treatment outcomes including viral load, CD4 and body weight at baseline and after one year follow-up period. Analysis for statistical difference in treatment outcomes was performed using paired sample t-test. The observed occurrence of retention in care, presence of opportunistic infections and LTFU were reported in terms of numbers and percentages.

## Ethical consideration

This study utilizes program data from the IHVN cART program. Ethical Approval for the implementation of cART was obtained from the National Health Research Ethics Committee (NHREC). Also, approval for the study was acquired from the Institute of Human Virology Nigeria (IHVN). At the period of recruitment, all participants were formally recruited and provided consent to be decentralized into the community. To protect confidentiality, dataset was de-identified before retrieval from the database and dataset remained only accessible to the research team.

## Results

As illustrated in Fig 1, 251 eligible participants were recruited from 11 healthcare facilities located in the Federal Capital Territory (FCT) and Nasarawa states. These patients were then devolved into 51 CAGs. Table 1 provides a summary of demographic characteristics of the

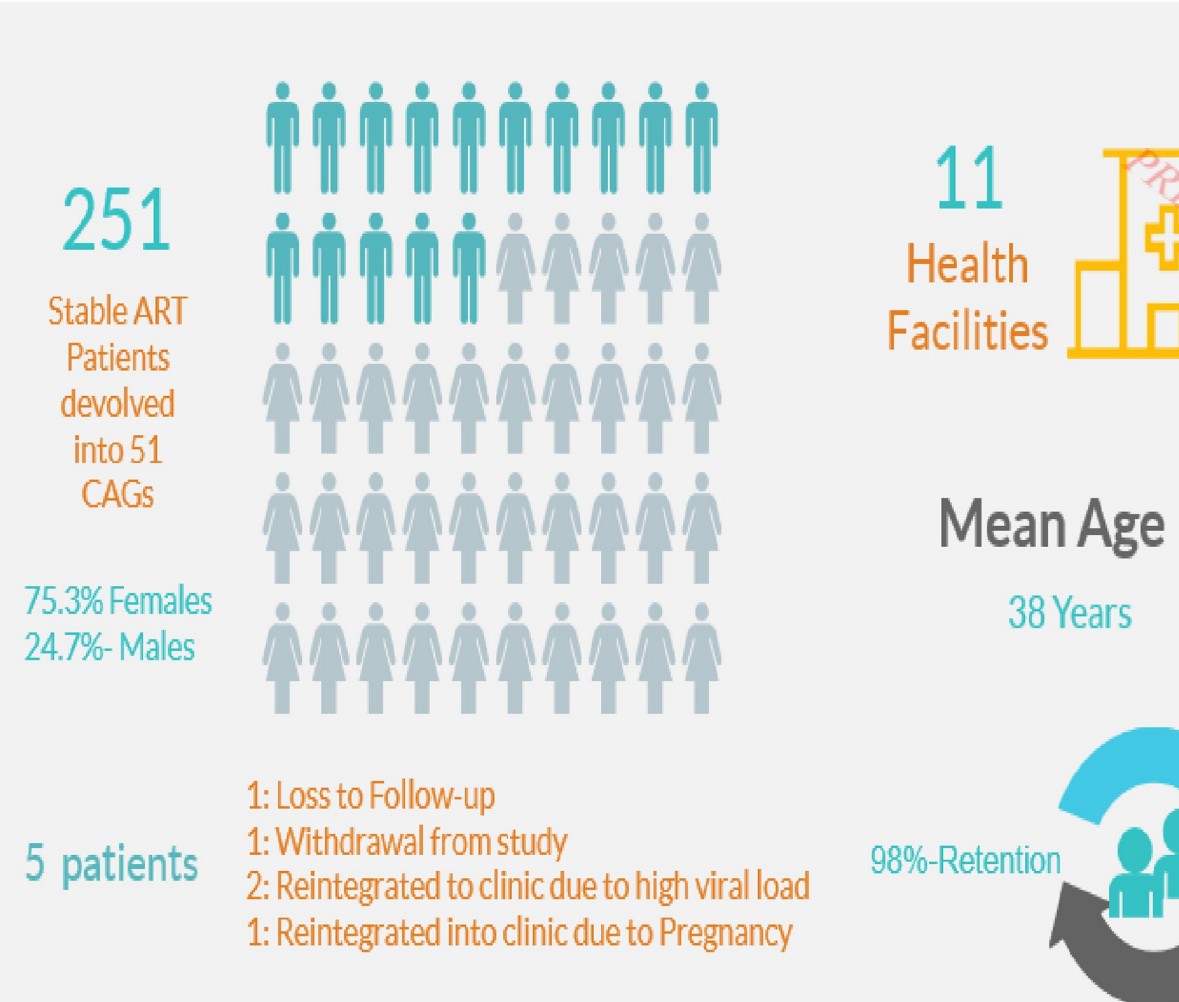

**Fig 1. Pictograph of community antiretroviral group selection outcomes.**

study participants. The mean age (standard deviation) of study participants was 38.0 years (9.5). Majority had completed secondary education [73(31.9%)], followed by primary [68 (29.7%)], and tertiary [52(22.7%)] education while only 36(15.7%) had no formal education completed. In addition, 46(19.41%) of the study participants were unemployed in comparison to 125(52.74%) participants who were self- employed and 66(27.85%) employed. The mean baseline CD4 counts and viral load ($log_{10}$) results were 456.4cells/mm$^3$ (288.9) and 3.5 c/ml (1.1) respectively.

A total of 246 (98.0%) CAG patients were retained after a year of follow-up (Fig 1). Out of 5 (2.0%) patients who were not retained in care, 1 participant was Lost To Follow Up (LTFU), 2 transferred to health facility due to high viral load, 1 transferred to the healthcare facility due to pregnancy and 1 withdrew from the study (Fig 1).

Fig 2 illustrates shows that after one year of follow-up, 165 out of 246 CAG patients who had blood samples collected for viral load test had their result available. Also, only 73 participants had CD4 results available after a year of follow up.

**Table 1. Characteristics of participants at baseline of community ART group.**

| Variable | Number(%) or Mean[S.D.] |
|---|---|
| **Age** | 38.0[9.5] |
| **Baseline CD4 counts (cells/mm³)** | 456.4[288.9] |
| **Log₁₀ baseline Viral Load (c/ml)** | 3.5[1.1] |
| **Baseline Body Weight(kg)** | 41.0[77.7] |
| **Sex** | |
| Male | 61(24.70) |
| Female | 186(75.30) |
| **Educational Level** | |
| None | 36(15.72) |
| Primary | 68(29.69) |
| Secondary | 73(31.88) |
| Tertiary | 52(22.71) |
| **Occupational Level** | |
| Unemployed | 46(19.41) |
| Employed | 66(27.85) |
| Self Employed | 125(52.74) |
| **Facility** | |
| General Hospital Karshi | 32(12.75) |
| General Hospital Kubwa | 10(3.98) |
| General Hospital Nyanya | 5(1.99) |
| National Institute for Pharmaceutical Research and Development (*NIPRD*), Abuja | 8(3.19) |
| *Asokoro District Hospital*, Abuja | 24(9.56) |
| General Hospital Bwari | 18(7.17) |
| Medical Centre, Mararaban Gurku | 69(27.49) |
| General Hospital Doma | 16(6.37) |
| Medical Primary Health Centre, Shabu | 56(22.31) |
| Dalhatu Araf Specialist Hospital, Lafia | 13(5.18) |

Note: % = percentage, S.D. = standard deviation, kg = kilogram, c/ml = copies per ml, cells/mm³ = cells per cubic millimeter.

155 out of 165 CAG patients with viral load results had remained virally suppressed after one year of follow-up (Fig 3). Additionally, 70 out 73 patients had CD4 counts >200 cells/ mm³ after one year of follow-up (Fig 3).

Table 2 below shows the results of paired sample T-test for CD4 counts and viral load before and a year after initiation of CAG. The mean (S.D.) of CD4 counts before and a year after initiation of CAG were 456.4 cells/mm³ (288.9) and 481.5 cells/mm³ (202.2). Although there was an increase in the mean CD4 counts after initiation into CAG with a mean difference of 25.1 cells/mm³, it was not statistically significant [*P*-value 0.5]. Also, the means (S.D.) viral load (log₁₀) results before and a year after commencing CAG were 3.5 c/ml (1.1) and 3.7c/ml (2.0). Although there was an increase in viral load with a mean difference of 0.15c/ml, it was not statistically significant [*P*-value *0.4*].

The mean (S.D.) of body weight at initiation and a year after CAG among study participants were 60.8kg (12.1) and 65.9kg (12.5) respectively. With a mean difference of 5.1kg, there was a statistically significant increase in the bodyweight of CAG clients after a year of commencing cART [*P*-value *0.01*].

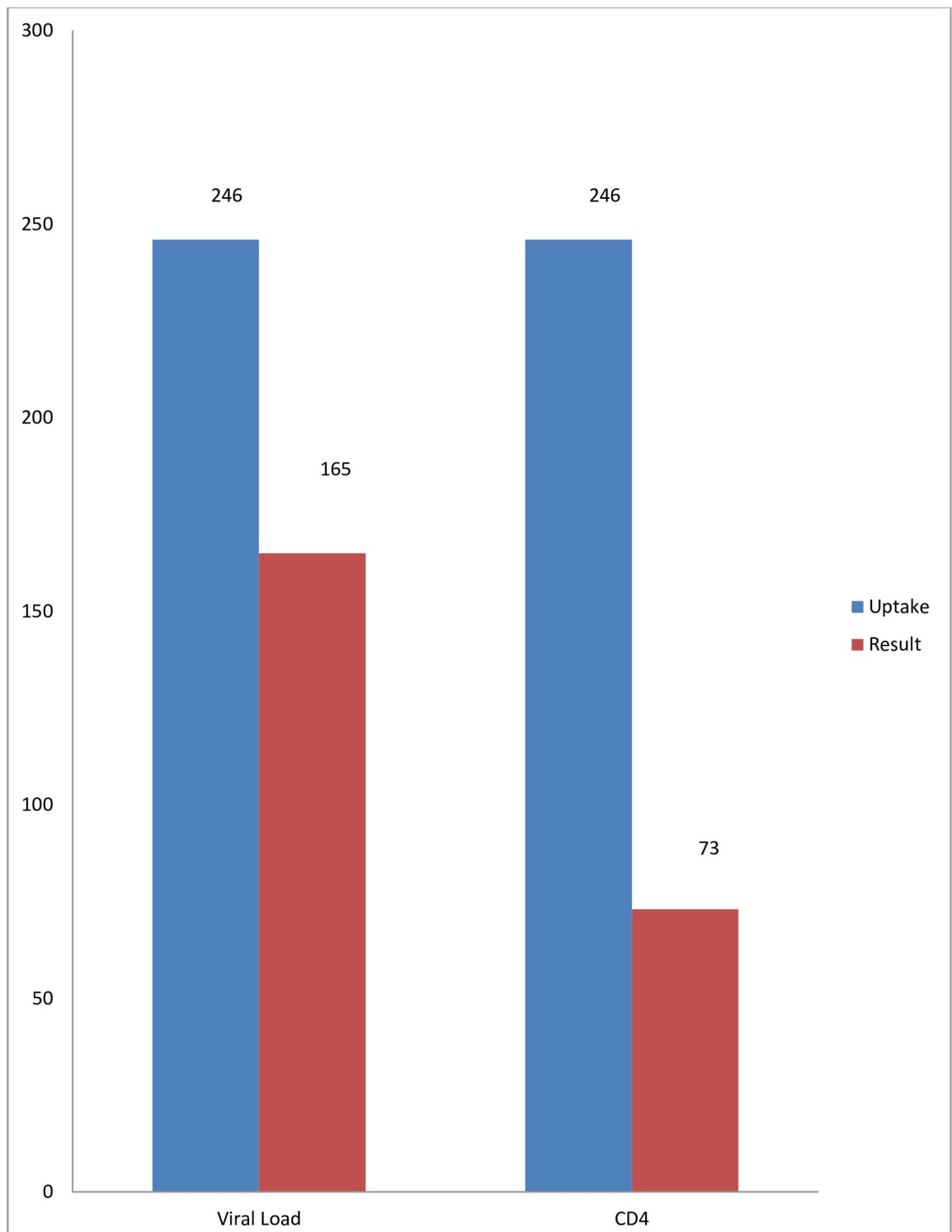

**Fig 2. Summary of test uptake against results after one year of follow-up.**

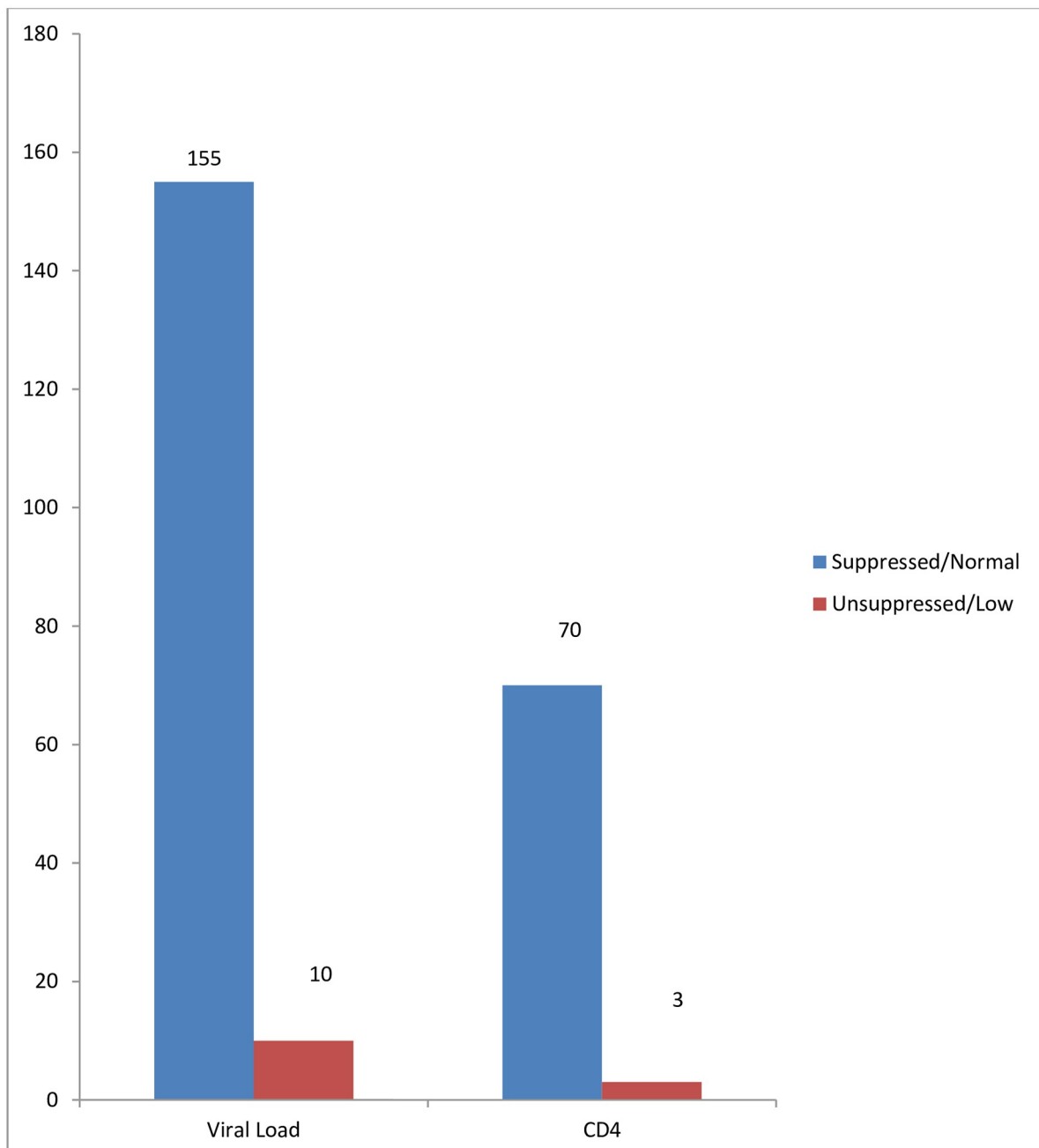

**Fig 3. Summary of test results after one year of follow-up.**

## Discussion

This study has demonstrated that CAG may be an effective model for maintaining optimum treatment outcomes while improving retention in care among stable ART patients. After a year follow-up of CAG participants, the virologic and immunologic health outcomes of the participants remained stable and satisfactory. Also, we observed that participants significantly gained body weight, and there was only minimal concern about retention in care after one year follow-up.

**Table 2. Differences in the participants' bodyweight, immunologic and virologic status after a year follow-up.**

| Variable | X | S.D. | S.E. | 95% C.I. | P-value |
|---|---|---|---|---|---|
| **CD4 Count** | | | | | 0.50 |
| Baseline CD4 (c/mm$^3$) | 456.4 | 288.9 | 37.9 | 380.4–532.3 | |
| Repeat CD4 (c/mm$^3$) | 481.5 | 202.2 | 26.6 | 428.3–534.6 | |
| **Viral Load (VL)** | | | | | 0.40 |
| Log$_{10}$ Baseline VL (c/ml) | 3.5 | 1.1 | 0.8 | 3.4–3.7 | |
| Log$_{10}$ Repeat VL (c/ml) | 3.7 | 2.0 | 0.2 | 3.4–4.0 | |
| **Body Weight (BW)** | | | | | 0.01 |
| Baseline BW | 60.8 | 12.9 | 0.8 | 20.0–130.0 | |
| Repeat BW | 65.9 | 12.5 | 0.8 | 39.0–134.0 | |

Note: % = percentage, S.D. = standard deviation, c/ml = copies per ml, cells/mm$^3$ = cells per cubic millimeter, VL = Viral load, S.E. = Standard error, C.I. = Confidence interval, B.W. = Bodyweight.

We observed that retention of CAG patients in care was highly encouraging based on routine clinical visits and participation in CAG activities during the course of one year follow-up. The proportion of patients retained in care after the observation period was very high considering the existing evidence from a systematic review which revealed expected rates of LFTU at one year ranging from 1.2% to 26% [35]. Studies have also shown higher retention in care among CAG patients in comparison with their counterparts who were receiving health facility-based care [28,36].

A plausible explanation for the high retention in care among patients in CAGs could be due to reduction in the cost of transportation, distance, multiple clinic attendance per year and long waiting time and this is consistent with other studies that have shown significant increase in adherence and retention in care rates among patients in CAGs [28,29,37–39]. However, it has been reported that as follow-up period increases, retention in care decreases [40,41]. The inverse relationship between the duration of follow-up of CAG patients and retention in care has been confirmed by other studies conducted in SSA [28,42].

As indicated in this study, retention in care was affected by loss-to-follow up, transfer to health facility due to high viral load, pregnancy and patient withdrawal from the CAG. This, finding is also consistent with findings from previous studies [40,42–44].

Besides observing high retention in care, the immunologic and virologic status of participants after one year follow-up period remained stable and satisfactory. Prior findings from similar settings have also found insignificant or improved changes on virologic and immunologic health status of ART patients after devolvement into CAGs [28,29,37–39]. This can be attributed to the potential of CAG in improving retention in care which may also improve adherence to treatment and quality of care among CAG members [28,29,37–39]. CAG model also ensures easy and timely access to ARVs by addressing the challenges experienced in healthcare facility-based HIV care [45].

Additionally, the stability observed in the virologic and immunologic status of participants could be explained by the continuous support and interaction within the CAGs which encourages social network between patients as they are expected to actively provide adherence and social support, monitor wellbeing and represent members of the group in drug pickups.

The significant increase in the average body weight of CAG patients can be explained by the unchanged virologic and immunologic health status after one year of devolvement into CAGs. Thus, the virologic and immunologic stability of the patients after one year follow-up indicates an improvement in quality of care, lower risk of disease progression and

opportunistic infections some of which are related to increased energy expenditure and may be complicated by the presence of vomiting, mal-absorption, anorexia and increased nutritional losses [46,47]. It is also important to note that while other studies may have reported deaths among CAG patients, none was observed in this study [28,43,45].

Although the model has been effective in improving viral load and CD4 count test uptake among CAG patients, the major limitation is the poor turnaround time of test results as presented in our data. This shows the limitation of the health system in monitoring treatment outcomes among ART patients and remains a major concern that needs to be addressed.

One of the key prospects of this model is its flexibility. This model allows for unforeseen circumstances where patients are irresponsive to treatment or are experiencing certain co-morbidities. In such situation, CAG patients are immediately reintegrated into the healthcare facility for adequate treatment and care.

This study is the first evaluating the implementation of cART model in Nigeria. As a result, it is highly relevant giving the potential of cART in addressing the overburdened health system in resource-limited settings. This ART model has also demonstrated notable prospects in encouraging ownership and improving sustainability of ART at the community level. Nevertheless, some limitations have been identified in this study. This study applied "pre and post" design to evaluate the effect of CAG on stable ART patients. As a result, this design cannot account for bias due to the time difference between baseline and endpoint. Also, only stable ART clients were eligible for inclusion into the model which introduced selection bias. Therefore it is important that the model is re-evaluated to achieve a wide coverage and more patients benefit from the prospect of the model. While this study examined the effect of CAG among stable ART patients, the validity of our findings is limited to only stable ART patients. Secondly, the sample size of this study was not pre-specified and may not be representative of the target population. However, the sample size was drawn from the healthcare facility where all ART patients are treated. Also, study outcomes were examined within the same patients with no inter-patient variability. Therefore we acknowledged that the variation between patient clusters was not considered in our analysis. It may also be argued that the length of follow-up in this study was not sufficient. However, this study is an on-going implementation of CAG model and will be reexamined after three and five-years follow-up periods. This will provide adequate data on the long-term effect of the model on ART patient's treatment outcomes.

Community-based ART programs have achieved remarkable results in expanding access to ART in resource-poor settings. These programs have also promoted retention in care and catalyzed efforts to build health systems that respond efficiently to the burden of HIV disease. CAG can also reduce the workload on the healthcare system and allow closer monitoring for clients that need critical care.

Findings from this study may have some implications on HIV program implementation. Given the numerous potential benefits of cART on improved treatments outcomes and relieving the burden on the health system, it will be beneficial to further research thee prospects of this treatment model in order to effectively scale up cART for stable patients especially in resource-limited settings where financial cost and distance are major limitations to HIV treatment and care. Scaling-up of cART may also contribute towards achieving the 2nd and 3rd 90s of the UNAIDS target. By adopting this model especially in resource-limited settings, this may help address some of the challenges limiting effective HIV treatment and care. While this study do not provide evidence of the potential of CAG model in the prevention of nosocomial infections, other studies have identified the effectiveness of this model in reducing hospital-based treatment [48,49]. Therefore it is important that future studies are conducted to evaluate the effects of CAG model on nosocomial infection Future studies should also assess the effectiveness of cART using a more rigorous epidemiological design. Though CAG shows

promising evidence in improving treatment outcomes among stable ART patients, it is also important to identify how to improve this model to accommodate unstable ART patients.

## Conclusion

In summary, changes in immunologic, virologic and clinical outcomes remained insignificant after one year of devolvement of stable ART patients into CAGs. It is important that further studies be conducted to determine the effects of CAGs in comparison with unexposed group (health facility-based clients) among stable ART patients. This will provide the government of Nigeria, HIV program implementers and donors, with vital information necessary for actions to scale-up CAG programs in Nigeria.

## Acknowledgments

The authors gratefully acknowledge the United States President's Emergency Plan for AIDS Relief (PEPFAR), Federal Capital Territory and Nasarawa State Health Management Board, Network of People Living with HIV in Nigeria (NEPWHAN) FCT and Nasarawa state chapters for their technical contributions to this study.

## Author Contributions

**Conceptualization:** Patrick Dakum, Gbenga A. Kayode, Asabe Gomwalk, Helen Omuh, Halima Ibrahim, Mercy Omozuafoh, Abimiku Alash'le, Charles Mensah, Young Oluokun.

**Data curation:** Juliet Ajav-Nyior, Timothy A. Attah, Gbenga A. Kayode, Franca Akolawole.

**Formal analysis:** Gbenga A. Kayode.

**Investigation:** Timothy A. Attah.

**Methodology:** Timothy A. Attah, Gbenga A. Kayode.

**Software:** Timothy A. Attah, Gbenga A. Kayode.

**Supervision:** Juliet Ajav-Nyior, Timothy A. Attah, Gbenga A. Kayode, Asabe Gomwalk, Helen Omuh, Halima Ibrahim, Abimiku Alash'le, Charles Mensah.

**Validation:** Timothy A. Attah, Gbenga A. Kayode.

**Writing – original draft:** Patrick Dakum, Juliet Ajav-Nyior, Timothy A. Attah, Gbenga A. Kayode, Asabe Gomwalk, Charles Mensah, Young Oluokun, Franca Akolawole.

**Writing – review & editing:** Patrick Dakum, Juliet Ajav-Nyior, Timothy A. Attah, Gbenga A. Kayode, Asabe Gomwalk, Helen Omuh, Halima Ibrahim, Mercy Omozuafoh, Abimiku Alash'le, Charles Mensah, Young Oluokun, Franca Akolawole.

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
