## [Decision Letter · Decision Letter 0]

9 Nov 2020

PONE-D-20-25822

Effect of community antiretroviral therapy on treatment outcomes among stable antiretroviral therapy patients in Nigeria: A quasi experimental design

PLOS ONE

Dear Mr Attah,

Thank you for submitting your manuscript to PLOS ONE. After careful consideration, we feel that it has merit but does not fully meet PLOS ONE’s publication criteria as it currently stands. Therefore, we invite you to submit a revised version of the manuscript that addresses the points raised during the review process.

We look forward to receiving your revised manuscript.

Kind regards,

Professor Kwasi Torpey, MD PhD MPH

Academic Editor

PLOS ONE

Journal Requirements:

3. Please provide additional details regarding participant consent.

In the ethics statement in the Methods and online submission information, please ensure that you have specified what type you obtained (for instance, written or verbal, and if verbal, how it was documented and witnessed).

If your study included minors, state whether you obtained consent from parents or guardians.

If the need for consent was waived by the ethics committee, please include this information.

Additional Editor Comments:

The manuscript titled " Effect of community antiretroviral therapy on treatment outcomes among stable antiretroviral therapy patients in Nigeria: A quasi experimental design" describes the experiences with the decentralization of ART to CAGs. This is an important element in improving access of to HIV treatment whilst addressing bottlenecks. The CAG approach has been shown as a promising best practice. Though the implementation of CAG will add to the knowledge base, the manuscript has some methodological flaws that needs to be addressed.

1. Design: The study is described as quasi-experimental study. This is incorrect as the study does not have a control/comparison arm. Furthermore, the study was not prospective. From the review, data was collected from patients records before and after transition to CAGs. This design flaw needs to be addressed as it affects comparability. What happened to the group that remained in the facility? What was their outcomes? Are their characteristics similar to the CAG group?

2. Secondary data analysis: The authors describe the study as a secondary data analysis. This conflicts with the quasi-experimental design. The study was more of an implementation science research with a retrospective record review.

3. Ethics: There is lack of clarity of the ethical process. The patient gave consent to be devolved in to the CAGs but it is unclear whether the patient gave consent for the study. Given that this is routinely collected data in a program, it is expected to be in the exempt category. It is not clear if this was the approval provided by the ethics authority. This needs to be adequately explained.

4. Duration of treatment: The authors state that patients who have been on treatment for at least 6months.However, they do not describe the distribution of patients and the duration on treatment. This is important for the treatment outcomes.

5.Introduction: Line 34 Antiretroviral vaccines (ARV) - This should be corrected. Not vaccines

6. Line 43/44: Starting regionally...... Rephrase sentence. Does not read well

7. Line 59_ CAG - described as self-informed. This is inaccurate. It is rather self-formed not informed. it should be corrected throughout the manuscript

8. Line 73/74- Retention in care...: Sentence unclear. Please revise

9. Results: Though the paper is presented as CAGs which is a cluster of 5, the results are presented at the individual level . For example the 5 patients who dropped off, how many CAGs did they belong to ?

10. Results: Only 73 CD4 out of 251 was taken. This is about 29%. What is the implication of this on the findings?

11. Weight: providing the weight gain without highlighting how long long patients have been on treatment as outlined in #4 is problematic

12.Line 306_ Correct thee to these

13. Line 311- Typo. Replace THIS with IT

14. Line 312- Revise sentence for clarity

15. CAG preventing nosocomial infection is not supported by the data provided .

Reviewers' comments:

Reviewer's Responses to Questions

**Comments to the Author**

1. Is the manuscript technically sound, and do the data support the conclusions?

Reviewer #1: Partly

2. Has the statistical analysis been performed appropriately and rigorously? 

Reviewer #1: Yes

3. Have the authors made all data underlying the findings in their manuscript fully available?

Reviewer #1: No

4. Is the manuscript presented in an intelligible fashion and written in standard English?

Reviewer #1: Yes

5. Review Comments to the Author

Reviewer #1: Dear authors

With interest I read “Effect of community antiretroviral therapy on treatment outcomes among stable antiretroviral therapy patients in Nigeria: A quasi experimental design”.

Please find my comments:

Major revisions:

• Please check spelling throughout

• Ethics: “secondary data” suggests that data were collected for the purpose of this study. This seems to contrast with the info that patients provided consent to participate in the study and the mentioning of “A quasi experimental design” in the title, which show the used of a prospective design, thus relying on data primarily collected for the purpose of the study.

• Results: please provide data on retention / attrition for all 1,151 patients (“1, 251 eligible participants were recruited from 11 healthcare facilities located in the Federal Capital Territory (FCT) and Nasarawa states. These patients were then devolved into 51 CAGs”), even if details on characteristics and virological or immunological outcomes are not available. Also add the % of those eligible that were enrolled in CAG.

• Table 1: something went wrong, with page numbers overlapping data shown in the table

• Discussion, line 268: “Although the model has been effective in improving test uptake among CAG patients” I missed data on improved test uptake (improved compared to ?) in the text of the results section. Either remove this statement or show data. Please also specify to which type of tests you are referring.

• Discussion, line 280 : “This study applied “pre and post” design to evaluate the effect of CAG on stable ART patients. As a result, this design cannot account for bias due to the time difference between baseline and endpoint.” is not clear. Please explain how the design could have biased the findings.

• Discussion, line 285: “However, we believe the sample size drawn from the study population is representative of the target population”. As you did not assess whether the study pop is similar to the target pop you cannot claim this.

• Discussion: other limitations include selection bias (of 1,251 only a minority was included in CAG) and lack of data for VL and CD4

• Discussion, line 298: do not add new results, such as “a greater percentage of the clients have 298 suppressed viral load with no opportunistic infection throughout the duration of the study. This could be as a result of peer adherence counseling services and follow up activities within the community. A collaborative approach was applied during the development and implementation of the CAG as both the healthcare providers and patients were actively engaged. We observed that the healthcare providers and patients were excited about the implementation of this program.”

• Figure 2 and 3: please use 2-dimensions instead of 3 dimensions for the bar charts.

Minor revisions:

Title: please replace “design” by “study”. I expected to find a protocol, not original research output.

Ethics: replace “formerly” by “formally”?

Intro, line 45, “new 45 infections” please specify: HIV, TB, ???

Intro, line 47, “eliminate”, please use “reduce”

Intro, line 56, remove the full stop in the middle of the sentence

Intro, 78-80: Please remove “It is estimated that 95% of HIV service delivery is health facility-based in low and middle income countries (30). In nearly all countries, delivery of HIV care in the initial phase of rapid scale-up has been based on a “one-size-fits-all”, clinic-based approach.” This sentence fits better in the beginning of an earlier paragraph, now starting at line 58.

Methods, line 115: please edit “CAG is self-informed groups” and use “are”

Methods, line 117: “The 4 key functions of CAG model includes” should be “include”

6. PLOS authors have the option to publish the peer review history of their article (what does this mean?). If published, this will include your full peer review and any attached files.

Reviewer #1: **Yes: **Tom Decroo

---

## [Author Response · Author response to Decision Letter 0]

31 Mar 2021

EDITOR’S COMMENTS 

1) If there are ethical or legal restrictions on sharing a de-identified data set, please explain them in detail (e.g., data contain potentially identifying or sensitive patient information) and who has imposed them (e.g., an ethics committee). Please also provide contact information for a data access committee, ethics committee, or other institutional body to which data requests may be sent.

Response: Thank you for this comment. The data remains the property of the Institute of Human Virology, Nigeria. Dr Patrick Dakum is the chief investigator of the study, whose e-mail contact is (pdakum@ihvnigeria.org). Inquiries on how to have access to this data are highly welcomed and encouraged. Data request should be sent to the chief investigator. Access to this data is subject to the decision of the scientific committee of the organization after reviewing the proposal of the proposed study.

2) If there are no restrictions, please upload the minimal anonymized data set necessary to replicate your study findings as either Supporting Information files or to a stable, public repository and provide us with the relevant URLs, DOIs, or accession numbers. Please see http://www.bmj.com/content/340/bmj.c181.long for guidelines on how to de-identify and prepare clinical data for publication. For a list of acceptable repositories, please see http://journals.plos.org/plosone/s/data-availability#loc-recommended-repositories. We will update your Data Availability statement on your behalf to reflect the information you provide.

 Response: Thank you for this comment. As HIV is a sensitive issue, there are restrictions on the data. The data remains the property of the Institute of Human Virology, Nigeria. Dr Patrick Dakum is the chief investigator of the study, whose e-mail contact is (pdakum@ihvnigeria.org). Inquiries on how to have access to this data are highly welcomed and encouraged. Data request should be sent to the chief investigator. Access to this data is subject to the decision of the scientific committee of the organization after reviewing the proposal of the proposed study.

3) Please provide additional details regarding participant consent.

In the ethics statement in the Methods and online submission information, please ensure that you have specified what type you obtained (for instance, written or verbal, and if verbal, how it was documented and witnessed). 

If your study included minors, state whether you obtained consent from parents or guardians.

If the need for consent was waived by the ethics committee, please include this information.

 Response: 

Thank you for this comment. This HIV program evaluation study analyzed routine data generate from the PEPFAR funded cART program. Every HIV patient provided written consent at the point of enrollment to care that their data could be used to improve the program implementation. In addition, the authors obtained ethical approval for this study from the National Health Research Ethics Committee ((NHREC/01/01/2007-18/12/2019C). In line with the HIV program implementation, only HIV clients who were willing to be decentralized into the community involved in the cART program.

 4) Design: The study is described as quasi-experimental study. This is incorrect as the study does not have a control/comparison arm. Furthermore, the study was not prospective. From the review, data was collected from patients records before and after transition to CAGs. This design flaw needs to be addressed as it affects comparability. What happened to the group that remained in the facility? What was their outcomes? Are their characteristics similar to the CAG group?

Response: Thank you for your comment. We agree with the review that this study did not have a different group of patients as comparison group. However, this applied “before and after study design” -a form of quasi-experimental study. I have provided different references below. 

Also, it was stated by the reviewers that our study was not a prospective study because data was collected from patients records before and after transition to CAGs. Although we used our routine medical record to evaluate, we will like to state that study was prospective in nature because we have already planned to evaluate the program before the patients were devolved to the community ART groups. All the groups were prospectively followed up before the evaluation was conducted.

5) Secondary data analysis: The authors describe the study as a secondary data analysis. This conflicts with the quasi-experimental design. The study was more of an implementation science research with a retrospective record review.

Response: 

Thank you for your comment. We have corrected this mistake. As we mentioned above, this is a quasi-experimental study that was conducted prospectively using routine data.

6) Ethics: There is lack of clarity of the ethical process. The patient gave consent to be devolved in to the CAGs but it is unclear whether the patient gave consent for the study. Given that this is routinely collected data in a program, it is expected to be in the exempt category. It is not clear if this was the approval provided by the ethics authority. This needs to be adequately explained.

Response: Thank you for this comment. This HIV program evaluation study analyzed routine data generate from the PEPFAR funded cART program. Every HIV patient provided written consent at the point of enrollment to care that their data could be used to improve the program implementation. In addition, the authors obtained ethical approval for this study from the National Health Research Ethics Committee ((NHREC/01/01/2007-18/12/2019C). In line with the HIV program implementation, only HIV clients who were willing to be decentralized into the community involved in the cART program.

7) Duration of treatment: The authors state that patients who have been on treatment for at least 6months.However, they do not describe the distribution of patients and the duration on treatment. This is important for the treatment outcomes.

Response: Thank you for this comment. On page…7-8 line 106-108. We have provided information on the treatment duration as follows “cART was initiated in August 2017 and treatment outcomes were assessed after a one year follow-up period”. 

8).Introduction: Line 34 Antiretroviral vaccines (ARV) - This should be corrected. Not vaccines

Response: Thank you for this comment. This has been duly corrected.

9) Line 43/44: Starting regionally...... Rephrase sentence. Does not read well

Response: Thank you for this comment. This has been rephrased as follows “Regionally, there is a substantial deficit in the number of health workers. The Nigerian health workforce remains relatively below (8.7% clinician deficit) the recommended 23 clinicians per 1,000 population (11-13)”.

10) Line 59_ CAG - described as self-informed. This is inaccurate. It is rather self-formed not informed. it should be corrected throughout the manuscript

Response: Thank you for this comment. This has been corrected.

11) Line 73/74- Retention in care...: Sentence unclear. Please revise

Response: Thank you for this comment. The sentence has been corrected as follows “Improved retention in care and self reported adherence have also been identified among stable ART patients devolved into CAGs (27, 28)”.

12) Results: Though the paper is presented as CAGs which is a cluster of 5, the results are presented at the individual level. For example the 5 patients who dropped off, how many CAGs did they belong to?

Response: Thank you for this comment. As the review rightly mentioned, the analysis of this was at the individual level because of the number of participants was inadequate to assess the clustering effect. The participants that dropped off were from four CAGs. However, we do not think this information is relevant as the analysis was based at the individual level.

13) Results: Only 73 CD4 out of 251 was taken. This is about 29%. What is the implication of this on the findings?

Response: Thank you for this comment. On page 14 line 270-274the implication of this has been described as follows “Although the model has been effective in improving viral load and CD4 count test uptake among CAG patients, the major limitation is the poor turnaround time of test results as presented in our data. This shows the limitation of the health system in monitoring treatment outcomes among ART patients and remains a major concern that needs to be addressed”.

14) Weight: providing the weight gain without highlighting how long long patients have been on treatment as outlined in #4 is problematic

Response: Thank you for this comment. We did not capture this information because we used the same patient as a comparator of himself/herself. In other words, we did not capture this information because our study design has addressed inter-patient variability. Not adjusting for this factor (duration on treatment) has no impact on the validity of our findings. However, we agree with the reviewer that this information will be needed for the proper generalization of our results.

15) Line 306_ Correct thee to these

Response: Thank you for this comment. This has been corrected

16) Line 311- Typo. Replace THIS with IT

Response: Thank you for this comment. This has been corrected

17) Line 312- Revise sentence for clarity

Response: Thank you for this comment. The sentence has been revised as follows “While this study do not provide evidence of the potential of CAG model in the prevention of nosocomial infections, other studies have identified the effectiveness of this model in reducing hospital-based treatment (48, 49)”. 

18) CAG preventing nosocomial infection is not supported by the data provided.

Response: Thank you for this comment. We did not examine nosocomial infection as one of our outcomes. We mentioned it in the discussion to let the readers know that this study might have prevented some cases of nosocomial infections, as observed in other studies. We have cited an article to support this statement.

FIRST REVIEWER’S COMMENTS

1) Please check spelling throughout

Response: Thank you for this comment. Spellings have been duly corrected

2) Ethics: “secondary data” suggests that data were collected for the purpose of this study. This seems to contrast with the info that patients provided consent to participate in the study and the mentioning of “A quasi experimental design” in the title, which show the used of a prospective design, thus relying on data primarily collected for the purpose of the study.

Response: Thank you for this comment. This HIV program evaluation study analyzed routine data generate from the PEPFAR funded cART program. Every HIV patient provided written consent at the point of enrollment to care that their data could be used to improve the program implementation. In addition, the authors obtained ethical approval for this study from the National Health Research Ethics Committee ((NHREC/01/01/2007-18/12/2019C). In line with the HIV program implementation, only HIV clients who were willing to be decentralized into the community involved in the cART Although will used our routine medical record to evaluate, we will like to state the that study was prospective in nature because we have already planned to evaluate the program before the patients were devolved to the community ART groups. All the groups were prospectively followed up before the evaluation was conducted.

3) Results: please provide data on retention / attrition for all 1,151 patients (“1, 251 eligible participants were recruited from 11 healthcare facilities located in the Federal Capital Territory (FCT) and Nasarawa states. These patients were then devolved into 51 CAGs”), even if details on characteristics and virological or immunological outcomes are not available. Also add the % of those eligible that were enrolled in CAG.

Response : Thank you for your comment. This study only involved 251 participants. I believe the review misinterpreted the statement on page 10, line 161 where we stated as follows “As illustrated in figure 1, 251 eligible participants were recruited from 11 healthcare facilities located in the Federal Capital Territory (FCT) and Nasarawa states.” This statement was explaining that as shown in figure 1, the number of eligible participants recruited was 251. We did not recruit 1,251 participants. We have rephrased the statement as follows “Figure 1 shows that 251 eligible participants were recruited from 11 healthcare facilities located in the Federal Capital Territory (FCT) and Nasarawa states”.

4) Table 1: something went wrong, with page numbers overlapping data shown in the table

Response: Thank you for this comment. All the tables have been positioned appropriately.

5) Discussion, line 268: “Although the model has been effective in improving test uptake among CAG patients” I missed data on improved test uptake (improved compared to ?) in the text of the results section. Either remove this statement or show data. Please also specify to which type of tests you are referring.

Response: Thank you for this comment. This statement has been removed as it was not part of core objective of this study.

6) Discussion, line 280 : “This study applied “pre and post” design to evaluate the effect of CAG on stable ART patients. As a result, this design cannot account for bias due to the time difference between baseline and endpoint.” is not clear. Please explain how the design could have biased the findings.

Response: Thank you for this comment. Given that the parameters used as the comparators were measured at the baseline, it is possible that the extraneous factors that affected it at the baseline might be different from the extraneous factors that influenced the outcome at the end of the study. In other words, outcome assessment in a two-arm comparative study is expected to be carried out at the same time, but this is not the case in before and after study like ours. 

7) Discussion, line 285: “However, we believe the sample size drawn from the study population is representative of the target population”. As you did not assess whether the study pop is similar to the target pop you cannot claim this.

Response: Thank you for this comment. This statement has been corrected.

8) Discussion: other limitations include selection bias (of 1,251 only a minority was included in CAG) and lack of data for VL and CD4

Response: Thank you for this comment. On page 16, line 286-288, we have added more limitations of the study as follows “Also, only stable ART clients were eligible for inclusion into the model which introduced selection bias. Therefore it is important that the model is re-evaluated to achieve a wide coverage and more patients benefit from the prospect of the model”.

9) Discussion, line 298: do not add new results, such as “a greater percentage of the clients have 298 suppressed viral load with no opportunistic infection throughout the duration of the study. This could be as a result of peer adherence counseling services and follow up activities within the community. A collaborative approach was applied during the development and implementation of the CAG as both the healthcare providers and patients were actively engaged. We observed that the healthcare providers and patients were excited about the implementation of this program.”

Response: Thank you for this comment. This statement has been removed as you suggested.

10) Figure 2 and 3: please use 2-dimensions instead of 3 dimensions for the bar charts.

Response: Thank you for this comment. This has been updated

11) Title: please replace “design” by “study”. I expected to find a protocol, not original research output.

Response: Thank you for this comment. This has been corrected.

12) Ethics: replace “formerly” by “formally”?

Response: Thank you for this comment. This has been corrected

13) Intro, line 45, “new 45 infections” please specify: HIV, TB, ???

Response: Thank you for this comment. On page 5, line 46-48 we have rephrased the sentence as follows “There is also an increased risk of new HIV infections among health care workers of which health support staff are considered to be most vulnerable (12-17)”.

14) Intro, line 47, “eliminate”, please use “reduce”

Response: Thank you for this comment. This has been changed

15) Intro, line 56, remove the full stop in the middle of the sentence

Response: Thank you for this comment. This has been removed

16) Intro, 78-80: Please remove “It is estimated that 95% of HIV service delivery is health facility-based in low and middle income countries (30). In nearly all countries, delivery of HIV care in the initial phase of rapid scale-up has been based on a “one-size-fits-all”, clinic-based approach.” This sentence fits better in the beginning of an earlier paragraph, now starting at line 58.

Response: Thank you for this comment. Changes have been made based on your recommendation. The statement has now been moved to page 5, line 58-60.

17) Methods, line 115: please edit “CAG is self-informed groups” and use “are”

Response: Thank you for this comment. This has been corrected

18) Methods, line 117: “The 4 key functions of CAG model includes” should be “include”

Response: Thank you for this comment. This has been corrected

. 

Reference

1. Harris AD, McGregor JC, Perencevich EN, Furuno JP, Zhu J, Peterson DE, et al. The use and interpretation of quasi-experimental studies in medical informatics. J Am Med Inform Assoc. 2006;13(1):16–23.

2. Elder A. Research methodologies guide: Quasi-experimental design [Internet]. Iowa State university. 2008 [cited 2020 Nov 11]. Available from: https://instr.iastate.libguides.com/c.php?g=49332&p=318076

3. White H, Sabarwal S. Quasi-Experimental Design and Methods [Internet]. Unicef-irc.orgirc.org. [cited 2020 Nov 11]. Available from: irc.org
irc.org/KM/IE/img/downloads/Quasi-Experimental_Design_and_Methods_ENG.pdf

4. SAGE. Quasi Experimental and Single Case Experimental Design [Internet]. Sagepub.com. 2019 [cited 2020 Nov 11]. Available from: https://us.sagepub.com/sites/default/files/upm-binaries/89876_Chapter_13_Quasi_Experimental_and_Single_Case_Designs.pdf

5. University of Houston. Quasi-Experimental Designs [Internet]. Available from: https://uh.edu/~jmwillia/Methods_Cozby11.pdf

---

## [Editor Report · Decision Letter 1]

6 Apr 2021

Effect of community antiretroviral therapy on treatment outcomes among stable antiretroviral therapy patients in Nigeria: A quasi experimental Study

PONE-D-20-25822R1

Dear Mr. Timothy Attah,

We’re pleased to inform you that your manuscript has been judged scientifically suitable for publication and will be formally accepted for publication once it meets all outstanding technical requirements.

Kind regards,

Professor Kwasi Torpey, MD PhD MPH

Academic Editor

PLOS ONE

Additional Editor Comments (optional):

Comments have been addressed
---

## [Editor Report · Acceptance letter]

13 Apr 2021

PONE-D-20-25822R1 

Effect of community antiretroviral therapy on treatment outcomes among stable antiretroviral therapy patients in Nigeria: A quasi experimental study 

Dear Dr. Attah:

I'm pleased to inform you that your manuscript has been deemed suitable for publication in PLOS ONE. Congratulations! Your manuscript is now with our production department. 

Kind regards, 

on behalf of

Professor Kwasi Torpey 

Academic Editor

PLOS ONE